# Toward an Anthropometric Pattern in Elite Male Handball

**DOI:** 10.3390/ijerph19052839

**Published:** 2022-02-28

**Authors:** Florin Valentin Leuciuc, Ileana Petrariu, Gheorghe Pricop, Dan Mihai Rohozneanu, Ileana Monica Popovici

**Affiliations:** 1Department of Physical Education and Sport, Stefan cel Mare University of Suceava, 13 University Street, 720229 Suceava, Romania; ileana.petrariu@usm.ro (I.P.); gheorghe.pricop@usm.ro (G.P.); 2The Interdisciplinary Research Center for Human Motricity and Health Sciences, 13 Universitatii Street, 720229 Suceava, Romania; 3Department of Collective Games, Babes Bolyai University of Cluj-Napoca, 1 Mihail Kogalniceanu Street, 400084 Cluj-Napoca, Romania; dan.rohozneanu@ubbcluj.ro; 4Department of Physical Education and Sport, Alexandru Ioan Cuza University of Iasi, 30 Toma Cozma Street, 700554 Iasi, Romania; ileana.popovici@uaic.ro

**Keywords:** handball, pattern, anthropometry, performance

## Abstract

We investigated the anthropometric characteristics associated with specific handball skills in competition. The body anthropometric profiles differ significantly among the playing positions in handball due to the specific tasks. The aim of this study is to identify the anthropometric patterns for each playing position by collecting data from elite male handball players. To determine the anthropometric profile of the elite handball players for each playing position, we used descriptive statistics for every indicator in order to identify the optimal patterns for elite handball players from the top-four ranked teams at the most important competitions over a period of 18 years (2004–2021). Over time, the anthropometric indices evolved: the average height increased (from 190 to 192.6 cm) but less than weight increased (from 90.5 to 95.28 kg), and these affected the body mass index (increase from 25.2 to 25.67). The novelty of our study is that we identified an anthropometric pattern for each playing position and for all teams in elite male handball. Our study also covered a period of 18 years to give our results more accuracy and reliability.

## 1. Introduction

Currently, along with good sport preparation for elite athletes, the specific anthropometry indices are essential in order to achieve performance in top competitions in collective sports where others are also important: maturity status [1], capacity to be trained [2], body composition [3,4,5], somatotype [6,7,8], physiological performance characteristics [9], specific skills [10,11], playing position [12,13,14] and specifics of the sport branches [8,15,16,17,18].

In elite sport, the anthropometric indices have a major impact on performance [19,20,21,22] and additionally, there are other factors, such as the preparation level [23,24,25], specific preparation [26,27,28] and competition experience [29,30], that influence elite sport outcomes.

Handball has a complex character due to the acyclic movements and situations that appear asking players performing tasks according to the playing position both in attack and in defense. The modern handball game requires players to perform a high number of short, high-intensity specific actions. Based on these considerations, we can state that the features of anthropometric and motor particularities have a high degree of individualization according to the playing position, and it is interesting to determine how personal predispositions can compensate or substitute particular requirements of the playing position.

Somatic and skills indicators of the players can be essential in achieving certain tasks within the game, as in other game situations, they can be a barrier having a limited effect [31,32]. The anthropometric characteristics are determinant in order to efficiently apply the specific handball skills in competitions [33].

Body anthropometric profiles differ significantly among the playing positions in handball due to the specific tasks. From the anthropometric point of view, the wing players were found to be lighter (79.7–81.7 kg) and shorter (177.1–178.3 cm) than backcourt players (193.5–201.9 cm; 90.7–102.3 kg), goalkeepers (192.5–197.6 cm; 82.3–88.1 kg) and pivots (188.6–201.7 cm; 105–121.4 kg); pivots were heavier than centers (83.6–90.5 kg); backcourt players and pivots had higher muscular mass than wings; backcourt players had higher hand-grip values; and the line players (pivots) were the heaviest players.

The majority of elite handball players are part of the mesomorph and endomorph somatotypes; however, according to the playing positions, the backcourt players are mesomorph, wings and pivots showed an endomorph–mesomorph somatotype, and goalkeepers were in the ecto-endomorph somatotype zone [32,34,35]. The players having a higher skill level are usually taller, and their level of fat-free mass is higher, meaning more muscular mass and a better physiological level are required in modern elite handball in order to achieve performance [36,37,38].

From 1960 and 1970, there was information concerning the anthropometric characteristics of the elite handball players for each playing position, and, at least for weight, there are some differences compared to the current requirements [39]. In the study conducted by Taborsky, they presented data on anthropometric indices for players participating in the handball World Championships (W.C.) and Olympic Games (O.G.) in the 1970s and for W.C. and European Championships (E.C.) for the period 1998–2007. There is a visible but slight increase of height through the years, from an average of 184 cm in the 1970s to over 190 cm after the year 2000 [31].

A study concerning the anthropometric indices at senior handball teams participating in the competition E.H.F. Champions League, Final Four 2012 showed average values higher than in previous periods (height—192.48 cm, weight—94.63 kg and body mass index—25.51); however, there were only a small number of handball players (62) compared to O.G., W.C. and E.C., and it is important to mention that all of them are elite players of the best four teams in Europe and possibly of the world [40]. Another study conducted at W.C. in 2013 indicated the following values for anthropometric indicators for male handball players: height—190.10 ± 6.82 cm, weight—92.37 ± 9.80 kg and BMI—25.53 ± 2.09 [41].

Over time, there have been studies concerning physical characteristics of the male handball players with different numbers of participants and performance levels (national vs. elite) (Table 1).

In the studies with a small number of participants, the anthropometric data were collected directly by measuring handball players [42,43,44,45,46,47,48]. In the studies with a large number of subjects [41,49], the anthropometric data were obtained indirectly, without involving the researchers in the assessment process.

Based on the somatic requirements on high performance selection in handball, we can say that, in terms of height, only 25% of the population would meet the standard. The weight and the essential skills (coordination, speed and power) can compensate for a lower height [31]. The aim of this study is to identify the anthropometric patterns for each playing position by collecting data from elite male handball players.

## 2. Materials and Methods

### 2.1. Data Collection

This is a cross-sectional study where we use a descriptive analysis of the anthropometric indices of players at handball top competitions (E.C., W.C. and O.G.). The information concerning the anthropometric characteristics (weight and height) was collected from the EHF and IHF websites that present all information, including the team roster (age, anthropometric data, club and international matches). The data were obtained through an informative portal (EHF and IHF website; these two entities being the organizers of the competitions) without the necessary knowledge about the methodology of its collection in order to assess its reliability and validity [50,51,52,53,54,55,56,57,58,59,60,61,62,63,64].

### 2.2. Subjects

Our study collected data from 974 players participating at O.G., W.C. and E.C., as components of the teams ranked in the first four places at each competition. Each team had between 16 and 18 players listed for competition.

Inclusion criteria: male handball players from top-four ranked teams at O.G., W.C. and E.C. in the period 2004–2021. Exclusion criteria: female or male handball players from teams ranked out of the top four at O.G., W.C. and E.C. Due to the fact that this covers a period of 18 years, the data for the same player were used as long as he participated in these top handball competitions.

### 2.3. Statistics

Descriptive statistics were applied in order to identify the anthropometric patterns (height, weight and body mass index) for each playing position (wing, backcourt, center back, pivot and goalkeeper). The study collected information for almost 20 years (2004–2021). At O.G., there were 12 teams involved in the final stage; at W.C., 24 teams/32 teams (since 2021); and at E.C., the number rose from 16 to 24 teams (since 2020). The data were collected only from the players of the teams placed in the first four places at each analyzed competition.

To determine the anthropometric profile of the elite handball players for each playing position, we used descriptive statistics by applying the average, standard deviation, minimum value and maximum value for every indicator in order to identify the optimal patterns for elite handball players. We applied linear regression to find significant relationship between the variables used in our study using IBM SPSS Statistics 26. The coefficients for regression equation and tests of significance were the most important data obtained by applying this statistical method.

## 3. Results

Starting with O.G. 2004, data were collected for anthropometric characteristics of the male handball players who participated at O.G., W.C. and E.C. in order to identify the specific patterns for each playing position. We calculated the means for each parameter, standard deviation, minimum value and maximum value for each playing position and for all players. All collected data are presented in Table 2.

All the collected data allowed us to find useful information concerning anthropometric characteristics of the elite handball players for each playing position. The synthetic information for all analyzed competitions is presented in Table 3.

We applied linear regression to determine the significance of the collected data for the indicators used in our study for each playing position and for all players. The coefficients for regression used to identify the degree of significance are shown in Table 4. This statistical method was applied for a total of 18 situations. The statistical significance was achieved in 16 out of 18 situations: three for *p* < 0.05. 6 for *p* < 0.01, three for *p* < 0.001, and four for *p* < 0.0001. Statistical significance (Table 4) was not obtained only in two situations (both for goalkeepers).

In this context, we were able to identify the anthropometric pattern for each playing position, including all players as a whole by determining the lower and the upper limits for each of them (Table 5).

## 4. Discussion

The wings were shorter (average height—1.852 m, lower limit—1.80 m and upper limit—1.90 m) and also the lightest players (84.477 kg, minimum—79 kg and maximum—90 kg). The BMI average was 24.625, meaning they were normo-ponderal with a lower limit of 23.3 and an upper limit of 25.9. For all wing indicators, there were obtained statistical significance for *p* < 0.01 and *p* < 0.001 (Figure 1) [14,35,41,42,65,66].

Pivots (1.971 m average height, minimum—1.92 m and maximum—2.02 m) and backcourts (1.956 m, minimum—1.91 m and maximum—2.01 m) were the tallest and also the heaviest players, pivots—105.096 kg with accepted limits between 98 and 113 kg and backcourts—98.553 kg with optimal values between 92 and 105 kg. The average BMI for pivots was 27.207 and 25.588 for backcourts. For pivots, there was significance for *p* < 0.01 (height) and for *p* < 0.001 (weight and BMI) (Figure 1) [14,35,41,43,65,66].

Goalkeepers were close in value to pivots and backcourts (height average—1.944 m and accepted limits between 1.90 and 1.99 m, weight average—97.315 kg and optimal values between 90 and 104 kg and BMI average—25.745). At linear regression, we obtained statistical significance only for the BMI of goalkeepers (Figure 1) [14,35,45,48,65,66].

Backcourts were close in height to pivots (1.962 m), but lighter (98.553 kg) and the limits for BMI are between 24.2 and 27. The highest values of coefficients at linear regression were achieved at backcourts indicators (*p* < 0.0001) (Figure 1) [14,35,44,45,46,65,66].

Centerbacks are shorter than pivots, backcourts and goalkeepers but higher than wings with an average value of 1.903 m for height (limits between 1.85 and 1.95 m); the weight average was 92.697 kg (minimum—86 kg and maximum—99 kg); and the BMI average values was 25.609 (accepted range between 24.1 and 27.1). The level of significance for centerback indicators were for *p* < 0.01 (weight and BMI) and *p* < 0.001 (height) (Figure 1) [14,35,41,42,65,66].

The overall analysis of the pattern for a top handball player found that the average height was 1.926 m with optimal limits between 1.86 m and 2 m; the weight average was 95.28 kg with 86 to 105 kg accepted limits; and the average BMI was 25.677 with recommended limits between 24 and 27.3. We also obtained statistical significance for *p* < 0.001 (height) and *p* < 0.0001 (weight and BMI) (Figure 1) [14,47,48,49,67,68].

We consider the identified pattern in our study for each playing position valid as, in 16 out of 18 situations, we obtained statistical significance at linear regression, and the data were collected for a period of almost 20 years from 974 top male handball players (Figure 2).

Over time, the anthropometric pattern changes for each playing position compared to those in 1970 and 1980. The values for height and weight increased with an average value of 5%. The most important evolution trend was for pivots where the height increased by 7% and the weight by 20% [31,39].

Comparing the results to recent studies concerning the anthropometric pattern for top male handball players, there are no differences [32,35,40,67,68], and the players belonging to teams from the first half of the final ranking in top competitions are in the optimal limits for each indicator [41,58].

An interesting fact is that the pivots were the tallest (197 cm) and heaviest handball players (105 kg), as the backcourts (196 cm) are usually associated as the highest handball players. To identify the anthropometric pattern in our study, only players from the top four were included for each analyzed competition from the best male handball teams for period 2004–2021.

Every year, the I.H.F. awards the world’s best players; five goalkeepers, four centerbacks and two backcourts were nominated for the analyzed period. In that period of time, Nicola Karabatic and Mikkel Hansen were nominated three times. The world’s best players were in the limits of the anthropometric pattern identified in our study. The exception being the goalkeeper Arpad Sterbik, who exceeded the upper limits. This aspect confirms the validity of the anthropometric pattern determined by our study (Table 6).

## 5. Conclusions

The anthropometric pattern was found to evolve over time: the average height increased (1.4%) but less than the average weight (4.5%), thereby, influencing the average body mass index (2.7%). More weight in the case of elite handball players typically means more muscular mass required by the modern handball player in order to be effective in game actions during the competitions.

Anthropometric characteristics, including the body composition and the somatotype of the elite handball players, are specific to the playing position in order to allow them to efficiently act in competitions. Among these indicators, the specific preparation of handball players is essential to achieve competition goals and to be efficient in specific actions during the game.

The limitations of the study include access to data concerning only height and weight and indirect data collection from the official website of the International Handball Federation and European Federation. For the future, it will be important to collect data for other variables, such as the body composition, skinfold thickness and body circumference. Among other factors, the anthropometric pattern is essential when a coach wants to select a player for his team, and it is optimal to choose a player that meets these requirements concerning anthropometric patterns.

The novelty of our study is that we identified an anthropometric pattern for each playing position and for all teams in elite male handball, and our study covered a period of 18 years to give our results more accuracy and reliability. These identified patterns will be subject to change over time, and further studies are necessary in order to keep these anthropometric models updated.

## Figures and Tables

**Figure 1 ijerph-19-02839-f001:**
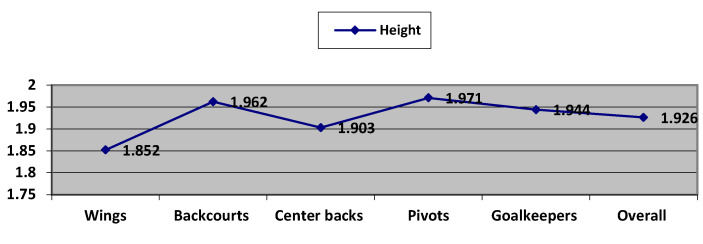
The average values for anthropometric indicators for each playing position.

**Figure 2 ijerph-19-02839-f002:**
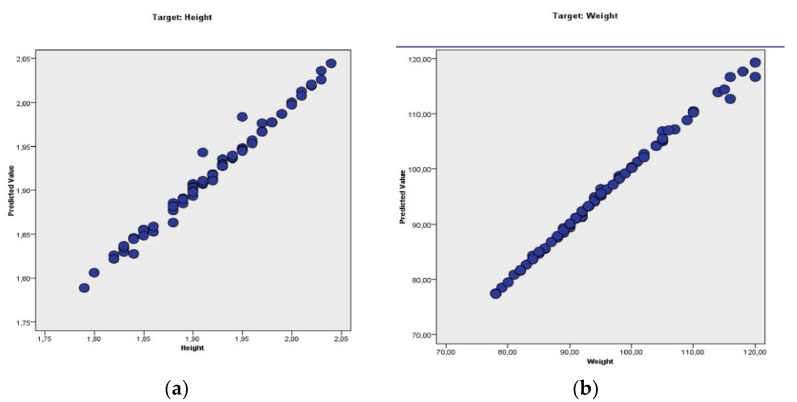
Linear regression model for anthropometric pattern indicators of the male handball players: (**a**)—height, (**b**)—weight and (**c**)—BMI.

**Table 1 ijerph-19-02839-t001:** The average values of the anthropometric indices according to the previous studies (updated after Ziv G., Lidor R., 2009).

Authors, Year	Level/No. of Participants	Height (cm)	Weight (kg)	BMI
Ghermănescu, Gogaltan, Jianu, Negulescu, 1983 [39]	Elite/-	188	86	24.36 ^1^
Bayios et al., 2001 [42]	National/15	181 ± 6	83.1 ± 5.2	25.41 ^1^
Gorostiaga et al., 2005 [43]	Elite/15	189 ± 8	95.2 ± 13	26.66 ^1^
Marques, Gonzalez-Badillo, 2006 [44]	Elite/16	184 ± 13	84.8 ± 13.1	25.01 ^1^
Asci, Acikada, 2007 [45]	National/16	185 ± 6	86.1 ± 8.9	25.17 ^1^
Marques et al., 2007 [46]	Elite/14	182 ± 7	82.5 ± 12.2	24.92 ^1^
Buchheit et al., 2009 [47]	National/9	181	78.4	23.97 ^1^
Sibila, Pori, 2009 [48]	National/78	188.44 ± 5.46	89.56 ± 8.41	25.23 ^1^
Leuciuc, 2012 [40]	Elite/62	192.48	94.63	25.51
Ghobadi, Rajabi, Farzad, Bayati, Jeffreys, 2013 [41]	Elite/409	190.10 ± 6.82	92.37 ± 9.80	25.53 ± 2.09
Michalsik, Madsen, Aagaard, 2015 [49]	National/157	188.7 ± 6.1	90.5 ± 7.9	25.42 ^1^
Pireva, 2019 [50]	National/133	186.84 ± 5.99	91.41 ± 10.31	26.19 ^1^

^1^ Data not presented in study and calculated by us.

**Table 2 ijerph-19-02839-t002:** The anthropometric indices for male handball players (2004–2021).

Competition	Playing Position /Anthropometric Indices/Statistics		Wings	Backcourts	Center Backs	Pivots	Goalkeepers	Overall
O.G. 2004	H(m)	X ± S	1.872 ± 0.033	1.995 ± 0.054	1.903 ± 0.035	1.994 ± 0.072	1.926 ± 0.043	1.940 ± 0.069
		MAX	1.92	2.11	1.95	2.14	2.00	2.14
		MIN	1.82	1.86	1.84	1.90	1.85	1.82
	W(kg)	X ± S	86.98 ± 4.833	100.83 ± 7.883	91.17 ± 4.104	109.20 ± 14.19	93.78 ± 3.456	95.64 ± 10.3
		MAX	95	118	98	132	100	132
		MIN	76	88	87	92	89	76
	BMI	X + S	24.84 ± 1.344	25.33 ± 1.741	25.19 ± 1.154	27.52 ± 3.539	25.29 ± 1.000	25.41 ± 1.995
		MAX	26.88	29.50	28.06	33.67	27.17	33.67
		MIN	22.63	21.79	23.91	22.55	24.16	21.79
O.G. 2008	H(m)	X ± S	1.834 ± 0.048	1.941 ± 0.026	1.899 ± 0.043	1.970 ± 0.045	1.950 ± 0.034	1.913 ± 0.066
		MAX	1.91	1.99	1.97	2.04	2.00	2.04
		MIN	1.76	1.90	1.85	1.90	1.91	1.76
	W(kg)	X ± S	83.86 ± 5.531	97.13 ± 6.019	92 ± 4.359	104.1 ± 6.008	95.88 ± 4.941	93.93 ± 9.146
		MAX	114	107	98	114	105	114
		MIN	95	85	86	97	90	85
	BMI	X ± S	24.94 ± 1.224	25.79 ± 1.350	25.52 ± 1.943	26.82 ± 1.245	25.21 ± 0.883	25.67 ± 1.367
		MAX	27.171	26.824	28.025	29.079	26.25	29.079
		MIN	23.735	23.546	22.933	25.250	23.467	22.933
W.C. 2009	H(m)	X ± S	1.816 ± 0.043	1.964 ± 0.047	1.898 ± 0.049	1.951 ± 0.043	1.954 ± 0.045	1.913 ± 0.076
		MAX	1.92	2.10	1.97	2.02	2.01	2.10
		MIN	1.73	1.89	1.81	1.89	1.90	1.73
	G(kg)	X ± S	79.06 ± 4.123	97.86 ± 4.704	92.00 ± 7.566	100.9 ± 6.619	97.63 ± 6.927	92.47 ± 9.917
		MAX	85	106	100	111	109	111
		MIN	72	85	78	91	90	72
	BMI	X ± S	23.97 ± 1.542	25.38 ± 1.041	25.54 ± 1.085	26.49 ± 1.545	25.58 ± 1.189	25.26 ± 1.459
		MAX	26.396	26.846	26.593	28.189	27.525	28.189
		MIN	21.267	23.129	23.809	23.212	23.467	21.267
W.C. 2011	H(m)	X ± S	1.856 ± 0.063	1.950 ± 0.029	1.895 ± 0.057	1.951 ± 0.040	1.951 ± 0.051	1.914 ± 0.064
		MAX	2.00	2.00	1.96	2.00	2.00	2.00
		MIN	1.76	1.91	1.78	1.87	1.86	1.76
	W(kg)	X ± S	83.17 ± 6.758	96.88 ± 4.559	91.36 ± 7.256	103.7 ± 5.461	98 ± 9.661	93.32 ± 9.770
		MAX	98	106	100	113	119	119
		MIN	73	92	80	98	91	73
	BMI	X ± S	24.14 ± 1.128	25.48 ± 1.251	25.43 ± 1.424	27.25 ± 1.078	25.73 ± 2.096	25.47 ± 1.662
		MAX	25.661	27.633	28.038	28.928	29.750	29.750
		MIN	22.531	23.232	22.819	25.252	23.467	22.531
O.G. 2012	H(m)	X ± S	1.854 ± 0.073	1.982 ± 0.062	1.924 ± 0.052	1.953 ± 0.043	1.929 ± 0.047	1.928 ± 0.075
		MAX	2.00	2.10	1.98	2.03	2.00	2.10
		MIN	1.78	1.89	1.83	1.87	1.85	1.78
	W(kg)	X ± S	88.25 ± 8.250	103 ± 7.458	95.43 ± 5.968	102.8 ± 5.412	95.38 ± 6.589	96.98 ± 9.123
		MAX	102	115	102	114	110	115
		MIN	75	93	83	93	90	75
	BMI	X ± S	25.68 ± 1.770	26.22 ± 1.097	25.77 ± 1.770	26.95 ± 0.897	25.64 ± 1.453	26.10 ± 1.292
		MAX	27.727	27.633	28.666	28.025	27.701	28.666
		MIN	23.148	24.984	22.992	24.967	23.467	22.992
W.C. 2013	H(m)	X ± S	1.847 ± 0.047	1.976 ± 0.077	1.911 ± 0.049	1.978 ± 0.034	1.950 ± 0.057	1.933 ± 0.079
		MAX	1.93	2.12	1.98	2.03	2.01	2.12
		MIN	1.78	1.84	1.84	1.92	1.86	1.78
	W(kg)	X ± S	83.65 ± 4.471	99.23 ± 6.436	89.63 ± 10.35	108.3 ± 5.610	99.13 ± 10.03	95.73 ± 11.17
		MAX	90	110	100	114	119	119
		MIN	75	90	74	100	90	74
	BMI	X ± S	24.52 ± 1.142	25.42 ± 1.631	24.54 ± 2.333	27.66 ± 1.369	26.07 ± 2.050	25.63 ± 1.941
		MAX	26.827	29.879	27.147	29.675	29.455	29.879
		MIN	21.914	22.472	20.074	25.252	23.514	20.074
E.C. 2014	H(m)	X ± S	1.829 ± 0.045	1.953 ± 0.055	1.920 ± 0.044	1.976 ± 0.039	1.942 ± 0.046	1.921 ± 0.072
		MAX	1.90	2.10	1.98	2.03	2.00	2.10
		MIN	1.78	1.84	1.86	1.92	1.88	1.78
	W(kg)	X ± S	82.63 ± 5.123	98.50 ± 7.288	92.44 ± 10.44	106.5 ± 6.072	96.33 ± 5.874	94.93 ± 10.49
		MAX	93	110	102	114	110	114
		MIN	75	85	74	99	92	74
	BMI	X ± S	24.71 ± 0.985	25.82 ± 1.667	25.08 ± 2.574	27.28 ± 1.346	25.54 ± 1.274	25.71 ± 1.775
		MAX	26.827	30.840	28.597	29.117	27.500	30.840
		MIN	23.148	23.669	20.074	25.252	23.750	20.074
W.C. 2015	H(m)	X ± S	1.832 ± 0.049	1.952 ± 0.0450	1.896 ± 0.064	1.991 ± 0.056	1.930 ± 0.032	1.917 ± 0.075
		MAX	1.92	2.03	1.98	2.08	2.00	2.08
		MIN	1.77	1.83	1.80	1.92	1.89	1.77
	W(kg)	X ± S	81.12 ± 6.314	96.96 ± 6.779	90.63 ± 8.088	106.60 ± 7.764	92.22 ± 3.528	92.84 ± 10.62
		MAX	97	107	102	120	100	120
		MIN	70	82	80	99	87	70
	BMI	X ± S	24.16 ± 1.434	25.44 ± 1.660	25.20 ± 0.946	26.88 ± 1.439	24.76 ± 0.739	25.25 ± 1.626
		MAX	26.870	28.782	26.551	29.117	26.035	29.117
		MIN	21.605	20.870	23.872	24.750	23.356	20.870
E.C. 2016	H(m)	X ± S	1.859 ± 0.053	1.974 ± 0.057	1.918 ± 0.046	1.961 ± 0.041	1.960 ± 0.049	1.936 ± 0.067
		MAX	1.96	2.10	1.98	2.04	2.02	2.10
		MIN	1.78	1.87	1.82	1.91	1.89	1.78
	W(kg)	X ± S	83 ± 3.305	99 ± 7.148	92 ± 4.837	105.92 ± 7.292	100 ± 11.95	95.82 ± 10.28
		MAX	90	115	100	121	119	121
		MIN	78	88	85	96	80	78
	BMI	X ± S	24.03 ± 1.525	25.41 ± 0.967	25 ± 0.927	27.55 ± 2.602	26.03 ± 2.371	25.56 ± 1.963
		MAX	26.841	27.803	27.171	33.168	29.750	33.168
		MIN	21.085	23.428	23.796	24.27	22.161	21.085
O.G. 2016	H(m)	X ± S	1.863 ± 0.044	1.972 ± 0.055	1.893 ± 0.043	1.993 ± 0.046	1.938 ± 0.048	1.936 ± 0.067
		MAX	1.94	2.10	1.94	2.07	2.01	2.10
		MIN	1.79	1.86	1.84	1.92	1.86	1.79
	W(kg)	X ± S	87.08 ± 4.071	101.53 ± 7.010	99.29 ± 7.566	107.30 ± 5.599	98.75 ± 5.445	98.47 ± 9.078
		MAX	94	115	106	118	110	118
		MIN	79	92	87	100	93	79
	BMI	X ± S	24.92 ± 0.916	26.11 ± 1.509	27.71 ± 1.488	27.01 ± 1.628	26.31 ± 1.403	26.26 ± 1.617
		MAX	26.57	30.35	29.99	29.84	28.06	30.35
		MIN	24.99	23.96	25.70	24.99	24.98	22.99
W.C. 2017	H(m)	X ± S	1.840 ± 0.044	1.962 ± 0.049	1.881 ± 0.073	1.976 ± 0.036	1.922 ± 0.045	1.923 ± 0.067
		MAX	1.92	2.04	1.97	2.02	2.01	2.04
		MIN	1.79	1.88	1.77	1.92	1.87	1.77
	W(kg)	X ± S	84.50 ± 5.798	97.17 ± 6.697	91.29 ± 9.673	107.75 ± 7.363	98.88 ± 7.039	95.80 ± 10.024
		MAX	93	110	107	115	112	115
		MIN	74	86	77	95	92	74
	BMI	X ± S	24.96 ± 1.304	25.24 ± 1.133	25.79 ± 1.460	27.59 ± 1.863	26.75 ± 1.658	25.90 ± 1.672
		MAX	26.841	27.633	27.853	29.839	30.068	30.068
		MIN	22.84	23.669	23.574	24.729	24.984	22.84
E.C. 2018	H(m)	X ± S	1.881 ± 0.066	1.943 ± 0.047	1.914 ± 0.040	1.965 ± 0.051	1.956 ± 0.054	1.930 ± 0.060
		MAX	2.02	2.03	1.96	2.05	2.02	2.05
		MIN	1.78	1.86	1.84	1.88	1.89	1.78
	W(kg)	X ± S	87.41 ± 5.501	96.83 ± 6.813	94.00 ± 4.497	104.62 ± 9.332	100.30 ± 7.056	96.11 ± 8.850
		MAX	100	112	104	116	115	116
		MIN	78	85	87	88	90	78
	BMI	X ± S	24.72 ± 1.415	25.63 ± 1.255	25.66 ± 1.156	27.10 ± 1.842	26.22 ± 1.837	25.79 ± 1.637
		MAX	28.09	28.058	28.06	29.839	29.115	29.839
		MIN	22.877	23.872	24.195	24.414	24.262	22.877
W.C. 2019	H(m)	X ± S	1.872 ± 0.054	1.966 ± 0.061	1.913 ± 0.035	1.974 ± 0.041	1.958 ± 0.045	1.935 ± 0.063
		MAX	1.98	2.12	1.96	2.03	2.01	2.12
		MIN	1.79	1.90	1.84	1.92	1.89	1.79
	W(kg)	X ± S	86.25 ± 4.740	98.26 ± 5.858	95.50 ± 6.474	104.64 ± 7.541	98.50 ± 5.043	96.32 ± 8.612
		MAX	97	115	109	115	105	115
		MIN	78	92	88	90	90	78
	BMI	X ± S	24.62 ± 0.829	25.43 ± 1.054	26.11 ± 1.947	26.87 ± 1.857	25.71 ± 0.772	25.72 ± 1.546
		MAX	25.96	27.33	30.84	29.94	26.78	30.84
		MIN	22.91	22.47	23.91	23.67	24.72	22.47
E.C. 2020	H(m)	X ± S	1.861 ± 0.048	1.965 ± 0.044	1.888 ± 0.060	1.961 ± 0.039	1.967 ± 0.067	1.928 ± 0.067
		MAX	1.93	2.03	1.97	2.02	2.06	2.06
		MIN	1.79	1.87	1.77	1.90	1.85	1.77
	W(kg)	X ± S	85.56 ± 4.56	98.38 ± 5.51	89.82 ± 6.29	105.25 ± 9.26	98.56 ± 7.44	95.26 ± 9.42
		MAX	96	110	99	120	110	120
		MIN	79	90	77	90	84	77
	BMI	X ± S	24.73 ± 1.23	25.47 ± 1.08	25.17 ± 0.95	27.35 ± 1.99	25.46 ± 0.91	25.58 ± 1.53
		MAX	27.45	28.06	27.17	29.68	27.14	29.68
		MIN	22.71	23.67	24.19	24.41	24.26	22.71
W.C. 2021	H(m)	X ± S	1.862 ± 0.048	1.942 ± 0.037	1.893 ± 0.033	1.970 ± 0.044	1.931 ± 0.066	1.921 ± 0.059
		MAX	1.97	2.02	1.96	2.04	2.03	2.04
		MIN	1.79	1.86	1.84	1.91	1.82	1.79
	W(kg)	X ± S	84.630 ± 4.609	96.731 ± 5.903	93.900 ± 4.977	107.867 ± 8.709	96.833 ± 9.581	95.634 ± 10.044
		MAX	92	110	105	120	120	120
		MIN	78	86	88	95	85	78
	BMI	X ± S	24.43 ± 1.15	25.65 ± 1.32	26.22 ± 1.50	27.78 ± 1.85	25.96 ± 1.99	25.87 ± 1.95
		MAX	27.17	28.63	29.71	31.80	31.56	31.56
		MIN	21.75	23.47	24.90	25.50	24.27	22.27

H—height. W—weight. BMI—body mass index. X—average. S—standard deviation. MAX—maximum value. MIN—minimum value.

**Table 3 ijerph-19-02839-t003:** Synthetic information for the anthropometric indices obtained in our study.

Playing Position/Anthropometric Indices/Statistics		Wings	Backcourts	Center Backs	Pivots	Goalkeepers	Team
H(m)	X ± S	1.852 ± 0.051	1.962 ± 0.050	1.903 ± 0.048	1.971 ± 0.046	1.944 ± 0.048	1.926 ± 0.068
	MAX	2.00	2.12	1.98	2.14	2.06	2.14
	MIN	1.76	1.83	1.77	1.87	1.85	1.76
W(kg)	X ± S	84.477 ± 5.199	98.553 ± 6.404	92.697 ± 6.830	105.096 ± 7.482	97.315 ± 6.953	95.280 ± 9.798
	MAX	114	118	106	132	120	132
	MIN	70	82	74	88	80	70
BMI	X ± S	24.625 ± 1.278	25.588 ± 1.317	25.609 ± 1.510	27.207 ± 1.762	25.745 ± 1.417	25.677 ± 1.647
	MAX	27.727	30.84	30.84	33.67	31.56	33.67
	MIN	21.085	20.87	20.074	22.55	22.161	20.074

**Table 4 ijerph-19-02839-t004:** The significance of the anthropometric patterns for male elite handball players by applying linear regression.

Playing Position/Anthropometric Indices/Statistics		Wings	Backcourts	Center Backs	Pivots	Goalkeepers	Team
Height (m)	t	3.02 **	6.17 ****	5.54 ***	3.98 **	2.08	3.73 **
*p*		0.009	0.0001	0.001	0.003	0.076	0.002
Weight (kg)	t	3.88 **	5.20 ***	3.50 **	3.08 **	1.74	6.18 ****
*p*		0.002	0.001	0.008	0.009	0.13	0.0001
BMI	t	4.01 ***	9.42 ****	3.27 *	2.88 *	2.82 *	8.30 ****
*p*		0.001	0.0001	0.011	0.016	0.026	0.0001

Significance for * *p* < 0.05. ** *p* < 0.01. *** *p* < 0.001. **** *p* < 0.0001.

**Table 5 ijerph-19-02839-t005:** Anthropometric patterns for male elite handball players.

Playing Position/Anthropometric Indices	Wings	Backcourts	Center Backs	Pivots	Goalkeepers	Team
Height (m)	1.80–1.90	1.91–2.01	1.85–1.95	1.92–2.02	1.90–1.99	1.86–2.00
Weight (kg)	79–90	92–105	86–99	98–113	90–104	86–105
BMI	23.3–25.9	24.2–27	24.1–27.1	25.5–29	24.3–27.2	24–27.3

**Table 6 ijerph-19-02839-t006:** Anthropometric characteristics for handball players designated by I.H.F. as world players of the year.

Year	Player	Playing Position	Anthropometric Data		
			Height	Weight	BMI
2004	Henning Fritz	Goalkeeper	1.89	90.5	25.34
2005	Árpád Sterbik	Goalkeeper	2	120	30.00
2006	Ivano Balić	Center back	1.90	96	26.59
2007	Nikola Karabatić	Center back	1.96	104	27.07
2008	Thierry Omeyer	Goalkeeper	1.92	93	25.23
2009	Sławomir Szmal	Goalkeeper	1.90	90	24.93
2010	Filip Jícha	Left back	2.01	105	25.99
2011	Mikkel Hansen	Left back	1.92	93	25.23
2012	Daniel Narcisse	Center back	1.89	93	26.04
2013	Domagoj Duvnjak	Center back	1.98	100	25.51
2014	Nikola Karabatić	Center back	1.96	104	27.07
2015	Mikkel Hansen	Left back	1.92	93	25.23
2016	Nikola Karabatić	Center back	1.96	104	27.07
2018	Mikkel Hansen	Left back	1.92	93	25.23
2019	Nikklas Landin	Goalkeeper	2.01	105	25.99

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
