# Peer review of "Toward an Anthropometric Pattern in Elite Male Handball"

_ijerph, 2022, doi:10.3390/ijerph19052839_

Round 1

Reviewer 1 Report

The article does not present the appropriate structure to be published in the IJERPH. It is a descriptive work in which the data has been obtained from different websites, so it is not possible to verify the procedures for obtaining anthropometrics measures, nor the technical error of measurement. On the other hand, most of the results are shown in the discussion section. In this section, a single bibliographical reference cannot be identified, since most of them are found in the introduction. Most of them, in the first paragraph of the text (30 references).

Author Response

Dear Reviewer,

Thank you for your valuable evaluation of our paper.

We used your recommendations in order to improve the quality of our article and also give us a fresh perspective to the topic that we approached.

There are the responses for each remark / suggestion.

The article does not present the appropriate structure to be published in the IJERPH. It is a descriptive work in which the data has been obtained from different websites, so it is not possible to verify the procedures for obtaining anthropometrics measures, nor the technical error of measurement.

We agree that it is a descriptive work and we collected data from websites, but these websites are only 2, first of EHF (European Handball Federation – the entity that manage handball activities and competitions in Europe) and IHF (International Handball Federation – the entity that manage handball activities and competitions for all World). So, these are specialized sites that provides specific information for handball game.

Concerning the procedures for obtaining anthropometric data that it is possible to encounter technical errors, but these data are provided by the staff of each participating team - that fact was mentioned in Materials and methods section.

It is hard to have access to a top team in handball, but for more top teams and for a long period of time it is almost impossible, and one way it is to collect data for organizers of these competitions (EHF, IHF) with the limitations concerning collected data that were mentioned in Conclusion section.

On the other hand, most of the results are shown in the discussion section.

To solve this issue, we moved results from Discussion section.

In this section, a single bibliographical reference cannot be identified, since most of them are found in the introduction. Most of them, in the first paragraph of the text (30 references).

We added new bibliographical references in Discussion section.

We agreed that are many bibliographical references at the beginning where we tried to emphases the role and the importance of the anthropometry in elite sport and especially in collective sports and also, we mentioned and other factors essentials for performance.

The novelty of our study it is that we identified an anthropometric pattern for each playing position and for all team in elite male handball and because our study covers a period of 18 years give to our results more accuracy and reliability.

We applied to special issue Handball: Sport and Health and we hope our paper will be read by handball specialists.

Reviewer 2 Report

Thank you for your submitted manuscript entitled, “Toward an anthropometric pattern in elite male handball’’. In this reviewer’s opinion, there are major issues that limit the scientific and practical relevance of the results obtained in the present study. Novelty of the study: What is the question being answered? What did the authors want to show? Relevance of the findings: I believe that this particular study does not have a great deal of practical impact because there is nothing new that would be unexpected or help the coach with training or testing. How did the authors to expect as a specific practical application?

ABSTRACT

  • The objectives of the study has not been sufficiently defined.
  • Clarify the subjects’ level and background
  • Why not use average of age, height, body mass and body mass index of participants
  • Could be a relevant conclusion of the present study to find what is important to know.

INTRODUCTION

  • Further, the part of introduction needs revision. A more conclusive use of appropriate literature can help here to clearly state the purpose of the study in order to develop hypotheses (or null hypotheses, whichever way is preferred). In turn, this will serve as the paper's framework as it will define the importance of findings, i.e. which results are being presented. In its present form, the paper states one quite unspecific hypothesis, therefore, results presented seem random which also influences the discussion. As a result, the section conclusion remains vague, or rather, conclusive take home messages cannot be offered to the reader.

METHOD

  • How was sample size determined? (Sampling technique!)
  • It is important that you help the reader with the context of the study concerning the subjects’ background, training status, time of year, day testing, nutrition, hydration, etc. and information that will allow other investigators to put your data into context with the literature.
  • What about the inclusion and exclusion criteria?
  • Make sure you have the proper informed consent statement in the paper, i.e., subjects were informed of the experimental risks and signed an informed consent document prior to the investigation.

RESULTS

  • The results are not clearly described. The rationales for variables in the study is needed. I do not understand what it means in the result section.

DISCUSSION

  • The discussion needs to reflect what you found, how it relates to the literature and each paragraph should be logical in sequence as at present it is a bit hard to follow. Make sure the paper’s importance and the need is clear to the reader.

CONCLUSION

  • Thus, my biggest concern is that the practical and cases of this particular study are not particularly high in the coach after reading this is not really thinking about changing any of their approaches to training or testing.
  • Why might one want to cite this paper? What is the true impact of the literature?

Author Response

Dear Reviewer,

Thank you for your valuable and professional remarks concerning our paper.

We used your recommendations in order to improve the quality of our article and also give us a fresh perspective to the topic that we approached.

There are the responses for each remark / suggestion.

. Novelty of the study: What is the question being answered?

What it is the anthropometric pattern for elite male handball players nowadays?

What did the authors want to show? Relevance of the findings: I believe that this particular study does not have a great deal of practical impact because there is nothing new that would be unexpected or help the coach with training or testing. How did the authors to expect as a specific practical application?

We wanted to identify the anthropometric pattern for each playing position and for all team based on collecting data for top 4 teams at each major competition (Olympic Games, World Championship, European Championship) for the period 2004-2021. There are another’s studies that deals the anthropometric pattern of the handball players, but using a small number of subjects. Our study identifies an anthropometric pattern for each playing position and for all team by using data collected during a period of 18 years and that give to our results more accuracy and reliability. These data are essential in the moment when one coach must choose between 2 close value players and one meet the anthropometric requirements and other no. Among other factors the anthropometric pattern it is essential when a coach want to select a player for his team and it is optimal to choose a player that meet these requirements concerning anthropometric pattern.

ABSTRACT

  • The objectives of the study has not been sufficiently defined.

We clearly defined the objectives of the study.

  • Clarify the subjects’ level and background

We mentioned the subjects’ level and background.

  • Why not use average of age, height, body mass and body mass index of participants

We mentioned these information in Materials and Methods section.

  • Could be a relevant conclusion of the present study to find what is important to know.

We added a relevant conclusion of our study.

INTRODUCTION

  • Further, the part of introduction needs revision. A more conclusive use of appropriate literature can help here to clearly state the purpose of the study in order to develop hypotheses (or null hypotheses, whichever way is preferred). In turn, this will serve as the paper's framework as it will define the importance of findings, i.e. which results are being presented. In its present form, the paper states one quite unspecific hypothesis, therefore, results presented seem random which also influences the discussion. As a result, the section conclusion remains vague, or rather, conclusive take home messages cannot be offered to the reader.

Previous studies present data for all teams and rarely for each playing position, collected for a small number of subjects (usually between 10 and 20).

In our study we wanted to collect data in order to identify the anthropometric pattern for each playing position in elite male handball.

We added data from previous studies concerning anthropometric pattern and we state clearly the aim of the study.

METHOD

  • How was sample size determined? (Sampling technique!)

Our study collected data from 974 players participating at O.G., W.C. and E.C., components of the teams ranked in the first 4 places at each competition. Each team had between 16 and 18 players listed for competition. Also we mentioned inclusion and exclusion criteria.

  • It is important that you help the reader with the context of the study concerning the subjects’ background, training status, time of year, day testing, nutrition, hydration, etc. and information that will allow other investigators to put your data into context with the literature.

The data have been obtained through an informative portal (EHF and IHF website; these 2 entities being the organizers of the competitions) without the necessary knowledge about the methodology of its collection in order to assess its reliability and validity. Data were collected in January (for European and World Championships) and in July (for Olympic Games) by the staff of each team.

  • What about the inclusion and exclusion criteria?

Were mentioned in this section.

  • Make sure you have the proper informed consent statement in the paper, i.e., subjects were informed of the experimental risks and signed an informed consent document prior to the investigation.

No need informed consent because data was collected by the organizers of the competitions.

RESULTS

  • The results are not clearly described. The rationales for variables in the study is needed. I do not understand what it means in the result section.

We added information be more clearly understand collected data.

DISCUSSION

  • The discussion needs to reflect what you found, how it relates to the literature and each paragraph should be logical in sequence as at present it is a bit hard to follow. Make sure the paper’s importance and the need is clear to the reader.

We made modifications and compared data with other studies concerning the topic that we approached.

CONCLUSION

  • Thus, my biggest concern is that the practical and cases of this particular study are not particularly high in the coach after reading this is not really thinking about changing any of their approaches to training or testing.
  • Why might one want to cite this paper? What is the true impact of the literature?

The novelty of our study it is that we identified an anthropometric pattern for each playing position and for all team in elite male handball and because our study covers a period of 18 years give to our results more accuracy and reliability.

We applied to special issue Handball: Sport and Health and we hope our paper will be read by handball specialists.

Reviewer 3 Report

The study is well structured, clear and contributes with interestingly insights with regard to the characterisation of elite handball players.
As mentioned by the authors, there are other variables as important or even more important for handball performance. 
Only table 2 seems too extensive. It would be desirable to try to present those results in a shorter way, for example complemented with figures?!

Author Response

Dear Reviewer,

Thank you for your valuable and professional remarks concerning our paper.

We used your recommendations in order to improve the quality of our article and also give us a fresh perspective to the topic that we approached.

There are the responses for each remark / suggestion.

The study is well structured, clear and contributes with interestingly insights with regard to the characterisation of elite handball players.
As mentioned by the authors, there are other variables as important or even more important for handball performance.

We agreed that are and others important variables for elite handball, but our study it was focused on the anthropometric pattern.

Only table 2 seems too extensive. It would be desirable to try to present those results in a shorter way, for example complemented with figures?!

In table 2 we wanted to present the statistics for each competition that we analyzed the anthropometric data. In table 3 it is presented only statistics for all analyzed period. There it is the possibility to eliminate table 2.

We used figures to represent the average values for each parameter of each playing position and also for linear regression model. 

Round 2

Reviewer 1 Report

I appreciate the effort made by the authors, but consider that the main problems of the article have not been solved.

Author Response

Dear Reviewer,

We want to respond point by point to your remarks and suggestions for our article in order to improve it and to be accepted for publishing in the IJERPH.

The article does not present the appropriate structure to be published in the IJERPH. It is a descriptive work in which the data has been obtained from different websites, so it is not possible to verify the procedures for obtaining anthropometrics measures, nor the technical error of measurement.

We agree that it is a descriptive work and we collected data from websites, but these websites are only 2, first of EHF (European Handball Federation – the entity that manage handball activities and competitions in Europe) and IHF (International Handball Federation – the entity that manage handball activities and competitions for all World). So, these are specialized sites that provides specific information for handball game. The data have been obtained through an informative portal without the necessary knowledge about the methodology of its collection in order to assess its reliability and validity. Data were collected in January (for European and World Championships) and in July (for Olympic Games) by the staff of each team.

Concerning the procedures for obtaining anthropometric data that it is possible to encounter technical errors, but these data are provided by the staff of each participating team - that fact was mentioned in Materials and methods section.

There are studies with a small number of participants where the anthropometric data were collecting directly by measuring handball players and other studies with a large number of subjects where the anthropometric data were obtained indirectly, without being involved the researchers in the assessment process (facts mentioned in our article).

It is hard to have access to a top team in handball, but for more top teams and for a long period of time it is almost impossible, and one way it is to collect data for organizers of these competitions (EHF, IHF) with the limitations concerning collected data that were mentioned in Conclusion section. We wanted to identify the anthropometric pattern for each playing position and for all team based on collecting data for top 4 teams at each major competition (Olympic Games, World Championship, European Championship) for the period 2004-2021. There are another’s studies that deals the anthropometric pattern of the handball players, but using a small number of subjects. Our study identifies an anthropometric pattern for each playing position and for all team by using data collected during a period of 18 years and that give to our results more accuracy and reliability. These data are essential in the moment when one coach must choose between 2 close value players and one meet the anthropometric requirements and other no. Among other factors the anthropometric pattern it is essential when a coach want to select a player for his team and it is optimal to choose a player that meet these requirements concerning anthropometric pattern.

Our study collected data from 974 players participating at O.G., W.C. and E.C., components of the teams ranked in the first 4 places at each competition. Each team had between 16 and 18 players listed for competition. Also we mentioned inclusion and exclusion criteria.

On the other hand, most of the results are shown in the discussion section.

To solve this issue, we moved results from Discussion section.

In this section, a single bibliographical reference cannot be identified, since most of them are found in the introduction. Most of them, in the first paragraph of the text (30 references).

We agreed that are many bibliographical references at the beginning where we tried to emphases the role and the importance of the anthropometry in elite sport and especially in collective sports and also, we mentioned and other factors essentials for performance.

We added new bibliographical references in Discussion section and compared our results to other studies that approach same topic.

Previous studies present data for all teams and rarely for each playing position, collected for a small number of subjects (usually between 10 and 20).

In our study we wanted to collect data in order to identify the anthropometric pattern for each playing position in elite male handball.

The novelty of our study it is that we identified an anthropometric pattern for each playing position and for all team in elite male handball and because our study covers a period of 18 years give to our results more accuracy and reliability.

We applied to special issue Handball: Sport and Health in the IJERPH journal and we hope our paper will be read by handball specialists.

Pleas mention if are other problems or suggestions that we need solve in order to improve the quality of our paper and to be accepted for publishing in the IJERPH.

Additionally, we mentioned that reviewers 2 and 3 gave their acceptance for publication after we made requested modifications in round 1.

Reviewer 2 Report

Amendments have been made against my comments and I am happy to endorse publication.  

Author Response

Dear Reviewer,

Thank you again for your remarks and suggestions that improved the quality of our paper and also for endorsing the publication.

Dear Reviewer,

Thank you again for your remarks and suggestions that improved the quality of our paper and also for endorsing the publication.